# Chitosan-Graft-Poly(acrylic acid) Superabsorbent’s Water Holding in Sandy Soils and Its Application in Agriculture

**DOI:** 10.3390/polym14235175

**Published:** 2022-11-28

**Authors:** Retno S. D. Lestari, Dhena Ria Barleany, Alia Badra Pitaloka, Meri Yulvianti, Dimas Prasetyo, Dendy Vito Anggoro, Adam Ruhiatna

**Affiliations:** 1Chemical Engineering Department, Universitas Sultan Ageng Tirtayasa, Jalan Jenderal Sudirman km. 3, Cilegon 42435, Indonesia; 2Magister of Chemical Engineering, Universitas Sultan Ageng Tirtaysa, Jalan Raya Jakarta km. 4, Serang 42124, Indonesia; 3Applied Biomaterial and Product Engineering Laboratory, Universitas Sultan Ageng Tirtayasa, Jalan Jenderal Sudirman km. 3, Cilegon 42435, Indonesia; 4Biomass Valorization Laboratory, Center of Excellent, Universitas Sultan Ageng Tirtayasa, Jalan Jenderal Sudirman km. 3, Cilegon 42435, Indonesia

**Keywords:** superabsorbent polymer, reusability, water holding, water retention

## Abstract

Sandy soil has a low ability to absorb and store low water, low nutrient content, and a high water evaporation rate, so it is not suitable to be used as agricultural land. Superabsorbents can be used to overcome these weaknesses. The purpose of this study was to measure the abilities of the superabsorbents, including water holding, swelling, and water retention of sandy soil enriched with superabsorbent, and to analyze the chitosan-graft-poly(acrylic acid) superabsorbent characteristics. The superabsorbent was prepared by mixing a chitosan solution with ammonium persulfate as an initiator and acrylic acid, which had been neutralized with KOH. Then, the mixture was cross-linked with *N*,*N*′-methylenebisacrylamide (MBA). The resulting superabsorbent gel was dried in an oven and then crushed for analysis. The results showed that an increase in chitosan concentration increased the gel fraction, swelling, reusability, and water holding. Meanwhile, an increase in chitosan concentration decreased water retention in sandy soils. The swelling kinetics can be predicted using the pseudo-second-order model with high accuracy (R^2^ value of 0.99).

## 1. Introduction

The conversion of agricultural land into industrial and residential areas has a direct impact on decreasing agricultural productivity. The limited land for agriculture has given rise to vertical and hydroponic farming methods. The disadvantages of these methods are that they require continuous and intensive care such as fertilization and watering and have a small production capacity which is not comparable to the growing demand for agricultural products. Sandy soil can be used to solve the problem of limited agricultural land. However, the problem is that the sandy soil has poor chemical and biological quality where the low organic matter content, total nitrogen, and ion exchange capacity result in low macro/micronutrient content, water retention, and soil fertility [1]. Because of these properties, there is no support for microbial activity in the sandy soil [2].

Increasing the function of sandy soil as agricultural land can be performed by adding a superabsorbent which has a simultaneous function of increasing water supply and allowing the release of fertilizer nutrients slowly. Superabsorbent is a three-dimensional cross-linked hydrophilic polymer that can absorb and hold large amounts of water and release it under stress conditions. Hydrophilic functional groups such as –OH, –CONH•, –CONH_2_, –COOH, and –SO_3_H function as water absorbers that attach to the polymer backbone and resist dissolution because of the cross-linking between networks [3,4,5,6].

Superabsorbents are usually made of synthetic polymers prepared by radical copolymerization, frontal copolymerization, graft copolymerization, cross-linking, and ionizing radiation [5]. The superabsorbent from synthetic polymers exhibits the advantages such as low cost, long service life, and high water absorption rate. However, it is dangerous for the environment because it cannot be decomposed in a reasonable time and disrupts plant growth [7,8,9]. To reduce the use of synthetic materials, this study combined a synthetic polymer with a natural polymer. The superabsorbent made of acrylic acid and chitosan was used as a carrier while retaining water in sandy soils.

The superabsorbent grafted with chitosan has the ability to absorb water up to hundreds of times its dry weight [10]. Superabsorbents from various synthetic polymers which can be grafted with chitosan are acrylic acid, acrylamide, and acrylonitrile [10,11,12,13]. Acrylic acid-co-acrylamide-based superabsorbents can also be grafted with chitosan [14]. Furthermore, the superabsorbent of chitosan-g-Poly(acrylamide)/Montmorillonite can be formed by grafting chitosan with acrylamide [15]. Meanwhile, the superabsorbent used for holding water in sandy soils is made by synthesizing acrylate, which is cross-linked with poly-vinylpyrrolidone (PVPP), using the solution polymerization method with tap water as a reaction medium [16]. El-Tohamy et al. [17] used a hydrogel superabsorbent made of poly(acrylate/acrylic acid) for water conservation in sandy soils.

The use of superabsorbent in sandy soil is to prevent water evaporation because it can effectively reduce the water evaporation rate to achieve the purpose of water retention. Therefore, we proposed to synthesize a superabsorbent from a mixture of synthetic and natural polymer (acrylic acid–chitosan), which was cross-linked with *N*,*N*′-methylenebisacrylamide (MBA). The chitosan-g-poly(acrylic acid) superabsorbent was tested for its ability to hold and retain water in sandy soils. In addition, the swelling was also measured and analyzed for the characterization of the resulting superabsorbent.

## 2. Materials and Methods

### 2.1. Materials

Acrylic acid (p.a., 99%; CAS Number: 79-10-7 from Sigma-Aldrich, St. Louis, MO, USA), chitosan (collected from PT. Biotech Surindo with degree of deacetylation (DD) of 87.2%), potassium hydroxide pellets for analysis EMSURE^®^ (CAS Number: 1310-58-3 from Merck KGaA, Darmstadt, Germany), *N*,*N*′-methylenebisacrylamide (MBA) (CAS Number: 110-26-9 from Sigma-Aldrich, St. Louis, MO, USA), ammonium persulfate (APS) (CAS Number: 7727-54-0 from Merck KGaA, Darmstadt, Germany), Nessler’s reagent B and potassium sodium tartrate tetrahydrate both from MERCK KGaA, Darmstadt, Germany were used in this study.

### 2.2. Preparation of Chitosan-Graft-Poly(acrylic acid) Superabsorbent

The preparation of the superabsorbent was started by dissolving 7.88 g of potassium hydroxide in 100 mL of distilled water. Next, the solution was used to neutralize 15 mL of acrylic acid and stirred for 15 min. The neutralized acrylic acid was then allowed to stand for 24 h to accelerate the process of gel formation after adding *N*,*N*′-methylenebisacrylamide (MBA). The next step was to make chitosan solutions with concentrations of 0.5%, 2%, and 4% *w*/*v*. Ammonium persulfate (0.05 g in 5 mL of distilled water) as an initiator was added to the chitosan solution. Then, it was mixed with the neutralized acrylic acid solution. The mixed solution was stirred at 500 rpm at a temperature of 70 °C for 5 min, then 0.1 g of N, N’-methylenebisacrylamide (MBA) was added, and the gel was stirred until a paste was formed. Chitosan-graft-poly(acrylic acid) superabsorbent was dried at 65 °C for 24 h or until the superabsorbent weight was constant. The dried superabsorbent was crushed for a series of analyses. Table 1 shows the symbols and descriptions for the variables used in this study.

### 2.3. Determination of Gel Fraction

Gel fraction analysis was carried out by modifying the analysis procedures reported by Erizal et al. [18]. The dried superabsorbent was pulverized, and as much as 0.5 g was put in a tea bag and then soaked in tap water for 24 h at room temperature. The hydrogen superabsorbent was dried again at 60 °C to a constant weight and then weighed. To determine the gel fraction, Equation (1) was used:(1)$$Gel\;fraction\;\left(\%\right)=\frac{W_{1}}{W_{0}}\times 100$$
where $W_{1}$ is the weight of the dried superabsorbent after immersion (g), and $W_{0}$ is the initial weight of the dried superabsorbent before immersion (g).

### 2.4. Measurement of Swelling and Kinetics

A total of 0.5 g of dried superabsorbent was put in a tea bag and then soaked in 200 mL of tap water at room temperature. The water absorption by the superabsorbent was measured at specific time intervals of 1, 3, 5, 12, and 24 h. Samples were taken, and the water absorption capacity was measured using Equation (2):(2)$$Swelling\;\left(g/g\right)=\frac{m_{1}-m_{0}}{m_{0}}$$

The swelling kinetics can be modeled using pseudo-first-order and pseudo-second-order kinetic models (Equations (3) and (4)). These swelling kinetic models were referred to in the studies by Chen et al. [19], Zhao et al. [20], Wang et al. [21], and Wang et al. [22]:(3)$$Q_{t}=Q_{e}\left(1-e^{-k_{1}t}\right)$$
(4)$$Q_{t}=\frac{k_{2}Q_{e}^{2}t}{1+k_{2}Q_{e}t}$$
where $Q_{e}$ and $Q_{t}$ are the absorption capacity at equilibrium and at time t, $k_{1}\left(\min^{-1}\right)$, and $k_{2}\left(g\cdot g^{-1}\;\min^{-1}\right)$ are kinetic constants of the two models, respectively.

### 2.5. Reusability

The reusability of SAP samples was measured using a method reported by Fang et al. [14] with the following procedures: The total of 0.5 g of the crushed sample was put in a tea bag and then soaked in 200 mL of distilled for 24 h. The water absorption capacity of the superabsorbents was measured using Equation (2). The swollen samples were dried in an oven at 65 °C to constant weights. The process of soaking and drying the superabsorbents was repeated about 4 to 5 times, measuring the water absorption capacity in each swelling condition.

### 2.6. Characteristics of Chitosan-Graft-Poly(acrylic acid) Superabsorbent

The spectra of chitosan-graft-poly(acrylic acid) superabsorbents were recorded using Bruker’s FT-IR spectrometer with wavenumbers ranging from 4000 to 500 cm^−1^. The morphology analysis was carried out using an SEM instrument (Evo 10-Carl Zeiss) coated with Au using a voltage of 10 kV. 

### 2.7. Water-Holding Capacity and Water Retention of Sandy Soils

The technique for determining the water holding and water retention of sandy soil was a modification from a technique reported by Matamedi et al. [23]. The superabsorbent was mixed with dried sandy soil with a mass ratio of the superabsorbent and dried sandy soil of 1:100 (*g*/*g*), and then the mixture was placed in a transparent container. Tap water was added slowly into the container to see the seepage from the bottom. The container was weighed before water was added $\left(W_{i}\right)$, and the container was weighed when the seepage of water was stopped from the bottom of the container $\left(W_{f}\right)$. The water-holding capacity (WHC %) of the sandy soil was calculated using Equation (5):(5)$$WHC\;\%=\frac{W_{f}-W_{i}}{W_{f}}\times 100$$

The water retention of sandy soils was determined using a method reported by Wang et al. [24]. A mixture of 100 g of sandy soil mixe and 1 g of chitosan-graft-poly (acrylic acid) superabsorbent was put in a tube with a nylon lining at the bottom and weighed $\left(W_{0}\right)$. Tap water was added to the sandy soil sample containing the superabsorbent slowly from the top. After no water seeped out of the tube, it was weighed again $\left(W_{1}\right)$. The tube was stored at room temperature and was weighed every 1 day $\left(W_{t}\right)$. Water retention of the sandy soil was calculated using Equation (6):(6)$$WR\;\left(\%\right)=\frac{W_{t}-W_{0}}{W_{1}-W_{0}}\times 100$$

## 3. Results and Discussion

### 3.1. Synthesis of Chitosan-Graft-Poly(acrylic acid) Superabsorbent

Synthesis of superabsorbent made of chitosan and acrylic acid using free radical graft depolymerization method. The proposed reaction mechanism of the superabsorbent polymer (SAP) synthesis process is shown in Figure 1.

The reaction mechanism that is proposed in Figure 1 begins with the decomposition of the ammonium persulfate initiator into sulfate anion radicals under heating conditions. The sulfate anion radical, which removes hydrogen from the –OH group of the chitosan backbone, results in the formation of a more active group [25,26]. This persulphate–saccharide redox system produces an active center which is the site of the reaction, where the monomer becomes the acceptor, and then the monomer molecule becomes a free radical donor to the nearest molecule, resulting in the growth of the polymer chain. At the same time, the polymer chains react with the vinyl groups of the MBA and form a superabsorbent polymer with a cross-linked structure [14,25,26].

### 3.2. Gel Fraction

Gel fraction is one of the parameters used to determine the efficiency of superabsorbent manufacture. The gel fraction is the fraction of the amount of starting material, either monomers or polymers, that are converted into superabsorbents. The gel fraction also shows the sensitivity of the material to the cross-linking process in converting it into a superabsorbent. The higher the sensitivity of the material is, the larger the gel fraction will be. The effect of chitosan concentration on the gel fraction can be seen in Figure 2.

The difference between superabsorbent 1 (SAP 1) with SAP 2 and SAP 3 was the concentration of chitosan added to acrylic acid in the superabsorbent manufacturing process. For SAP 1, SAP 2, and SAP 3, the concentrations of chitosan added were 4, 2, and 0.5 % (*w*/*v*), respectively. Figure 1 shows that SAP 1 and SAP 2 produced a larger percent gel fraction compared with SAP 3. The increase in chitosan concentration resulted in a higher percent of gel fraction. The use of chitosan in the manufacture of SAP was intended to reduce the use of synthetic polymers. In this study, the SAP manufacturing process was carried out using chemicals, such as ammonium persulfate as an initiator in the grafting process and *N*,*N*′-Methylenebisacrylamide (MBA) as a cross-linking agent.

The addition of a cross-linking agent caused the polymer chains to be cross-linked, and a gel-like material was obtained. At higher polymer concentrations, the macromolecules facilitating a cross-linking become closer. Increasing the concentration of chitosan (monomer) resulted in the availability of more active sites, increasing the rate of the polymerization reaction. In this study, SAP 1 and SAP 2 produced almost the same gel fraction; this means that the maximum gel fraction was 86%. If a higher chitosan concentration is used, the gel fraction may decrease. Erizal and Wikantan [27] stated that the increased concentration of chitosan could increase the gel fraction, but if the concentration continued to be increased, the gel fraction would decrease. The same was also reported by Saleem et al. [28], where a further increase in chitosan concentration resulted in a lower gel formation.

### 3.3. Measurements of Swelling Capacity and Kinetics

Water absorption is one of the key parameters in the manufacture of superabsorbents. The ability to absorb water is important because this superabsorbent is used as a medium to store water and release nutrients in the form of urea (nitrogen) in sandy soils that are low in nutrients. Figure 3 shows the effect of the concentration of chitosan on water absorption in the manufacture of superabsorbent from acrylic acid.

Figure 3 shows that increasing the chitosan concentration resulted in higher water absorption. SAP 1 with a chitosan concentration of 4% (*w*/*v*) had an absorption capacity of 159.92 g/g, SAP 2 with a chitosan concentration of 2% (*w*/*v*) had an absorption capacity of 139.72 g/g, and SAP 3 with a chitosan concentration of 0.5% (*w*/*v*) had an absorption capacity of 139.32 g/g. The addition of chitosan to the superabsorbent manufacturing process affected the network formed between cross-linked chitosan and acrylic acid chains with MBA. The decrease in chitosan concentration might cause the chain segments of the polymer chain to shorten with increasing chain cross-linking, reducing the swelling value of the SAP because of the limited space available for free water to enter the space of the cross-linked network [29].

The Swelling occurs because of the interaction between the functional groups of acrylic acid and chitosan with water through hydrogen bonds and because of the presence of a porous network in the superabsorbent. The use of chitosan had the effect of increasing water absorption because chitosan is a hydrophilic material that can absorb and hold liquids thousands of times their weight [30]. The increase in the concentration of chitosan was directly related to the increase in hydrophilicity which caused the water absorption also to increase [31]. The use of chitosan in the superabsorbent produces different effects; for example, a study by Erizal et al. [27] reported that an increase in the concentration of chitosan decreased the diffusion of water into the superabsorbent. Meanwhile, Saleem et al. [28] reported that increasing the concentration of chitosan increased the swelling value, but too high a chitosan concentration precisely reduced water absorption. This was due to the increase in viscosity because the increasing concentration of chitosan caused a steric effect to overcome the binding potential of the functional groups, resulting in a decrease in the expansion ratio. This study agreed with Saleem et al. [28] because the addition of chitosan could increase the hydrophilic properties of the superabsorbent.

The swelling kinetics of the superabsorbent of chitosan-graft-poly(acrylic acid) was determined using Equations (3) and (4). Figure 4 shows the comparison between the experimental data and calculated data obtained using the kinetic models (Equations (3) and (4)).

Figure 4 shows the comparison of calculated swelling data from the simulation using two swelling kinetic models, namely pseudo-first-order and pseudo-second-order models. The two kinetic models produced the best fitting with R^2^ values of 0.98–0.99. On the basis of the calculated data, the absorption capacity was in equilibrium at 180 min. While based on the experimental data, an increase was still observed in absorption capacity at that time, although it was not significant.

Table 2 shows the results of calculations to determine the value of absorption capacity at equilibrium (*Qe*) and rate constants (*k*) using the two kinetic models. The pseudo-first-order kinetic model assumes that the adsorption process originates from a physical process that occurred at the start of the adsorption in most cases. Meanwhile, the pseudo-second-order kinetic model is to predict behavior throughout the adsorption process, and it corresponds to the chemisorption mechanism as the rate controller [32].

Table 1 shows that the absorption of chitosan-graft-poly (acrylic acid) superabsorbent followed the pseudo-second-order kinetic model because it resulted in a higher R^2^ value than the pseudo-first-order kinetic model. This result indicated that the swelling process occurred in multistep. Initially, the water molecules were attracted by the hydrophilic groups and entered into the molecular cavity. Water molecules continued to fill the cavities until the internal and external osmotic pressures reached equilibrium [20]. The *Qe* value calculated using the pseudo-second-order model was close to the experimental data; this means that the swelling process of SAP 1, SAP 2, and SAP 3 showed that chemisorption was the rate-limiting step.

### 3.4. Superabsorbent Reuse

The reswelling ability of superabsorbent polymers is very important in their practical application. Figure 5 shows the results of reusing the superabsorbent after being used to absorb water, then drying and soaking it again in water. The process of water absorption or superabsorbent reuse was carried out 5 times.

Figure 5 shows the reswelling ability of acrylic acid–chitosan superabsorbent. With the increase in reswelling time, the water absorption of the superabsorbent polymer decreased. A repeated decrease in water absorption was caused by the gradual breakdown of the superabsorbent polymer network [33]. After swelling 5 times, these products still retained more than 60% water absorption capacity.

The excellent repeat swelling ability can increase the service life and reduce the cost of the superabsorbent polymer, which provides good application prospects for these products [14]. The excellent reusability can save costs with a long service life because the continuously porous structure of the chitosan-graft-poly(acrylic acid) superabsorbent provides excellent dimensional stability. Another advantage is that it can reduce the burden of environmental pollution [33].

### 3.5. Superabsorbent Characterization

#### 3.5.1. Scanning Electron Microscope (SEM)

Surface morphology analysis using SEM can be seen in Figure 6.

Figure 6 shows the morphology of acrylic acid–chitosan superabsorbent with a smooth surface. Figure 6A–D show the difference in SEM magnification. The superabsorbent’s surface morphology in Figure 6C,D with 1000× and 2000× magnification show the presence of pore and crack structures for fast penetration of water into the superabsorbent polymer particles. A connected visible pore structure serves to increase the water absorption capacity and the rate of water absorption. The same results of the morphological analysis were reported by Fang et al. [34], where the superabsorbent has a smooth surface and a pore structure. The interconnected pore structure can increase the water absorption capacity and the rate of water absorption [34].

#### 3.5.2. Fourier Transform Infrared Spectroscopy (FTIR)

Analysis of superabsorbent as a carrier for urea using FTIR can be seen in Figure 7.

The IR analysis for the chitosan-graft-poly(acrylic acid) superabsorbent shown in Figure 7 was supplemented by the IR analysis for chitosan (Figure 7A), which has been reported by Jayanudin et al. [35]. The IR analysis for chitosan supplemented in Figure 7 is intended to examine the peak change after grafting with acrylic acid and cross-linking with MBA. Figure 7B,C are IR for chitosan-graft-poly(acrylic acid) superabsorbent prepared using 0.5% *w*/*v* chitosan concentration (SAP 3) and 4% (*w*/*v*) chitosan concentration (SAP 1). Figure 7B,C show the FTIR spectra of SAP 3 and SAP 1, where the peaks produce almost the same wavelength.

The absorption bands detected at 3390 cm^−1^ (SAP 3) and 3425.58 cm^−1^ (SAP 1) were caused by the stretching vibration of O-H. Peaks at 2197.49 cm^−1^ (SAP 3) and 2919.6 cm^−1^ (SAP 1) were set for the C-H symmetric and asymmetric stretching vibrations. SAP 1 and SAP 3 had the bands at 1686.89 cm^−1^ and 1646 cm^−1^ showing the bands for the carbonyl stretching of amide I. The absorption band at 1555.4 cm^−1^ for SAP 3 and the absorption band at 1545.5 cm^−1^ for SAP 1 showed H symmetric and asymmetric stretching of –COO-. The characteristic absorption bands of N–H at 1598 cm^−1^ and 1380 cm^−1^ and C_3_–OH at 1094 cm^−1^ from raw chitosan could not be found because NH_2_, NHCO, and OH from chitosan took part in the graft reaction with acrylic acid [36]. Furthermore, the bands at 1440.29 cm^−1^ (C–H), 1402 cm^−1^, and 1030 cm^−1^ indicated the presence of poly (acrylic acid) chains. The absorption bands at 1164.89 cm^−1^ and 1164.4 cm^−1^ were associated with the C-N stretching vibration of MBA as a cross-linker [37].

### 3.6. Water-Holding Capacity and Water Retention of Sandy Soils

Analysis of water-holding capacity and water retention of sandy soil added with chitosan-graft-poly(acrylic acid) superabsorbent (SAP 1, SAP 2, and SAP 3) can be seen in Figure 8.

Sandy soil has weaknesses, such as very low organic matter, low water-holding capacity, high infiltration, low soil moisture, and very high soil surface temperature (26–40 °C). These factors cause sandy soils to be less suitable for plant growth [38]. To increase the productivity of sandy soil, which can later be used for agricultural land, an attractive solution is the addition of superabsorbent polymer to increase soil moisture because the polymer can hold water and reduce the rate of water evaporation.

The results showed that the addition of chitosan-graft-poly (acrylic acid) superabsorbent prepared using various chitosan concentrations could increase the water-holding capacity. The highest water-holding capacity (Figure 8B) was 67.4% with the addition of the SAP 1 polymer, which was prepared by grafting chitosan at a concentration of 4% (*w*/*v*). Almost the same result was obtained with the addition of SAP 2, resulting in a water-holding capacity of 67.18%. Meanwhile, the addition of SAP 3 resulted in a water-holding capacity of 62.56%. Increasing the concentration of chitosan successfully increased the water-holding capacity. The water-holding capacity of sandy soils without SAP addition was also measured, and the results were very low, only 18.23%. This research proved that the sandy soil, which was added with superabsorbent, could hold quite a large amount of water.

The ability of the superabsorbent made of chitosan and acrylic acid is that the –COO− group of the partially neutralized product can be converted into a –COOH group, and hydrogen bonds can be formed between –OH and –COOH (Figure 7) of the chitosan-based hydrogel. The –COOH group changes to the –COO− group, the hydrogen bond is separated, and the swelling capacity is increased because of the electrostatic repulsion of this carboxylic group as the main driving force. Furthermore, the cooperation of the sodium carboxylic group and the carboxylic acid in a suitable ratio when acrylic acid is partially neutralized can result in a higher water-absorbing ability [39]. The –COOH group has the capability of binding hydrogen to several water molecules per unit repeat. This makes the polymer swell and hold water firmly.

Figure 8B shows the water retention of sandy soil added with superabsorbent polymers (SAP 1, SAP 2, and SAP 3). Water retention indicated the rate of evaporation of water contained in the SAP. The slowest evaporation rate resulted from sandy soil with the addition of SAP 1. Sandy soil without the addition of SAP had a very fast evaporation rate; in only 2 days, the water content was lost, or the water retention was 0%. Meanwhile, after 7 days of incubation, water retention for SAP 1, SAP 2, and SAP 3 was 11.76%, 2.97%, and 0%, respectively. The ability to absorb water was due to the presence of hydrophilic groups that absorb water molecules that do not move in three-dimensional cross-links [40]. Water can be held at room temperature, which means that this SAP has good water-holding capacity. Chitosan-graft-poly(acrylic acid) superabsorbent can still retain water by 11.76% for 7 days, so it can be concluded that this SAP provides sufficient moisture for plants. This SAP can be a small reservoir to provide water for plants.

Figure 9 shows the ability of sandy soil to hold water before and after the addition of chitosan-graft-poly(acrylic acid) superabsorbent. Figure 9 shows that the sandy soil enriched with SAP can hold water, while without the addition of SAP, its ability to hold water was very low. Sandy soil with SAP can hold water so that the humidity of the sandy soil is maintained. SAP that was added to sandy soil had two simultaneous functions, namely to hold water and release nutrients if this SAP is used as a fertilizer carrier.

Applications of superabsorbents in agriculture have been widely carried out in land with low nutrient content to increase groundwater retention and in fertilizers to reduce evapotranspiration rates and increase soil aeration. The use of superabsorbent hydrogels directly affects soil permeability, density, structure, texture, evaporation, and water infiltration rate [41,42]. The addition of 0.3% superabsorbent significantly reduces the saturated hydraulic conductivity and increases the available water capacity [43].

## 4. Conclusions

The superabsorbent polymer of chitosan-graft-poly(acrylic acid) was synthesized through a free radical graft polymerization method with chitosan and acrylic acid as the skeleton structure and *N*,*N*′-methylenebisacrylamide (MBA) as a cross-linking agent. Characterization analysis for functional groups and microscopic surface morphology using FTIR and SEM confirmed that the chitosan-graft-poly(acrylic acid) superabsorbent polymer was successfully synthesized. The results showed that the best superabsorbent polymer resulting in maximum values in the analysis of gel fraction, swelling, and reusability was synthesized with the following conditions: 15 mL of acetic acid (99% *v*/*v*), chitosan concentration of 4% (*w*/*v*), ammonium persulfate concentration of 1% (*w*/*v*), and 0.1 g of *N*,*N*′-methylenebisacrylamide (MBA). The chitosan-graft-poly(acrylic acid) superabsorbent was then used to hold water, and water-retention test results showed that it can hold water up to 67.4% and still retain water in sandy soil for 7 days (water-retention analysis). In the swelling kinetic analysis, the pseudo-second-order kinetic model resulted in a very good R^2^ value of 0.99. These results indicate that the chitosan-graft-poly (acrylic acid) superabsorbent can be used to improve the function of sandy soil by increasing its absorption and enabling it to hold a large amount of water for a long time.

## Figures and Tables

**Figure 1 polymers-14-05175-f001:**
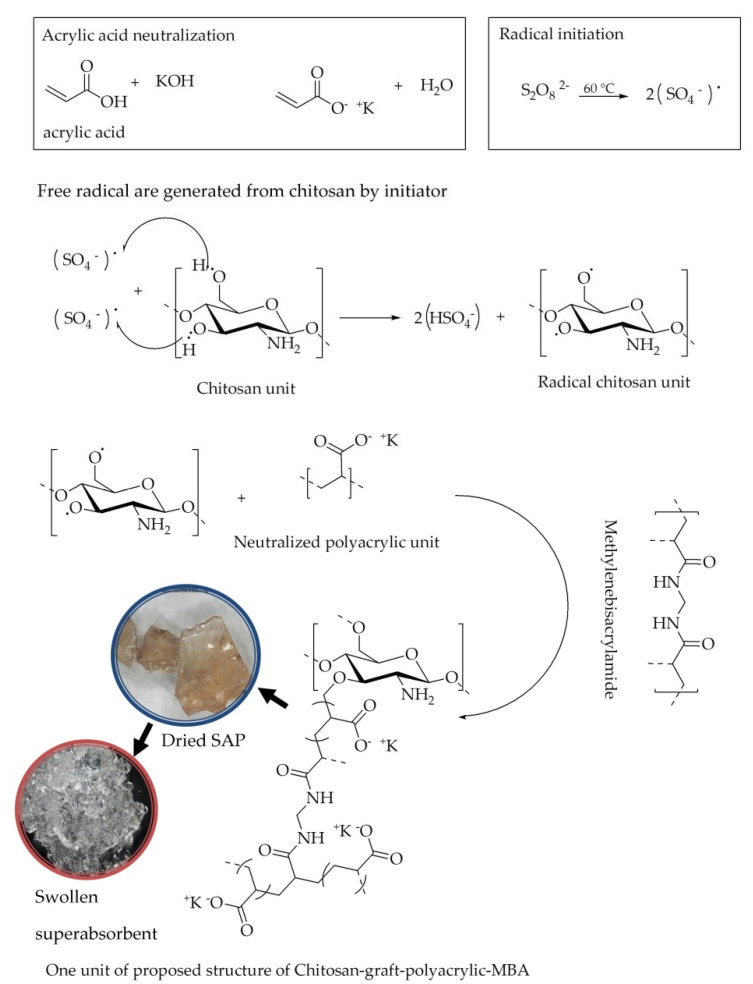
The proposed reaction mechanism of the synthesis process of graft polymerization of chitosan free radicals to acrylic acid with an MBA cross-linking agent.

**Figure 2 polymers-14-05175-f002:**
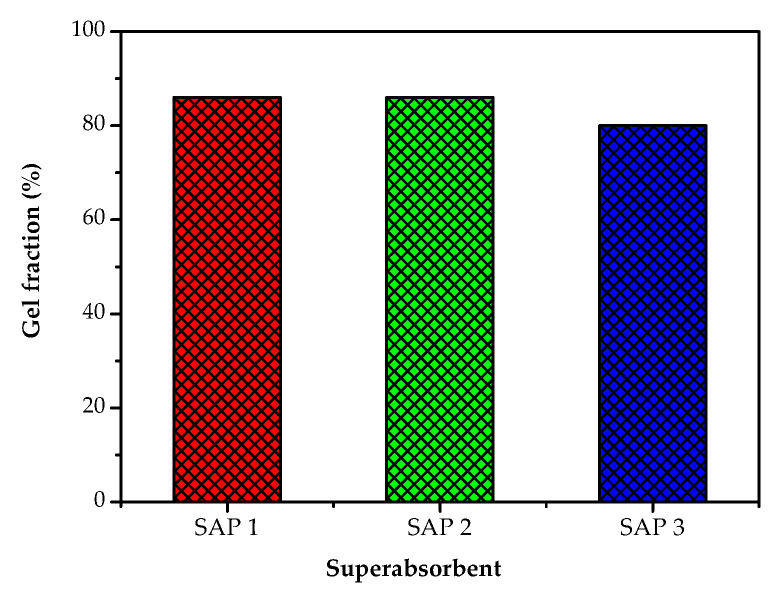
Effect of chitosan concentration in the manufacture of acrylic acid-based superabsorbent on gel fraction.

**Figure 3 polymers-14-05175-f003:**
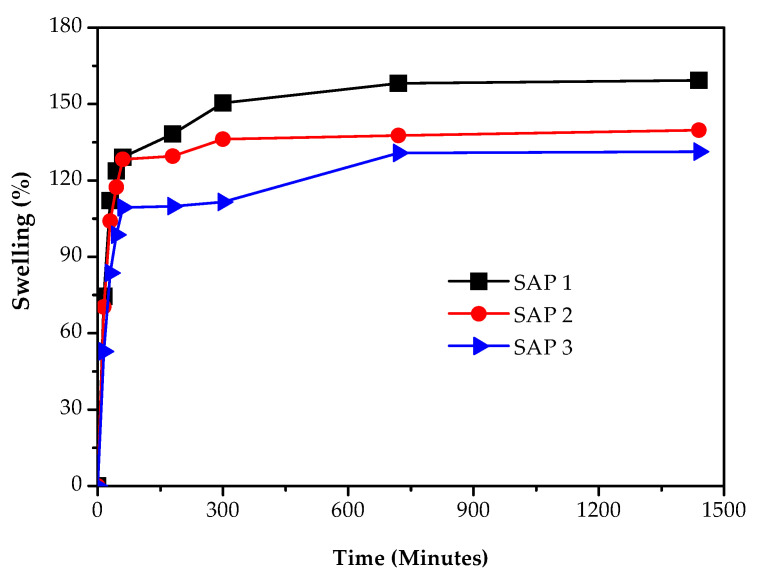
Effect of chitosan concentration on water absorption in manufacturing acrylic acid-based superabsorbent.

**Figure 4 polymers-14-05175-f004:**
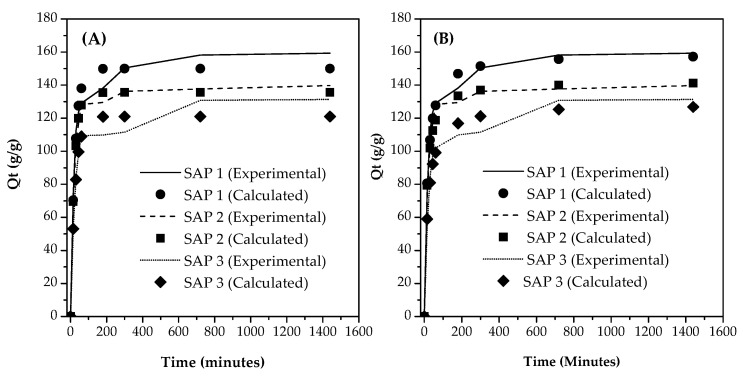
Comparison between experimental data and calculated data from the swelling kinetics model for (**A**) pseudo-first-order and (**B**) pseudo-second-order models.

**Figure 5 polymers-14-05175-f005:**
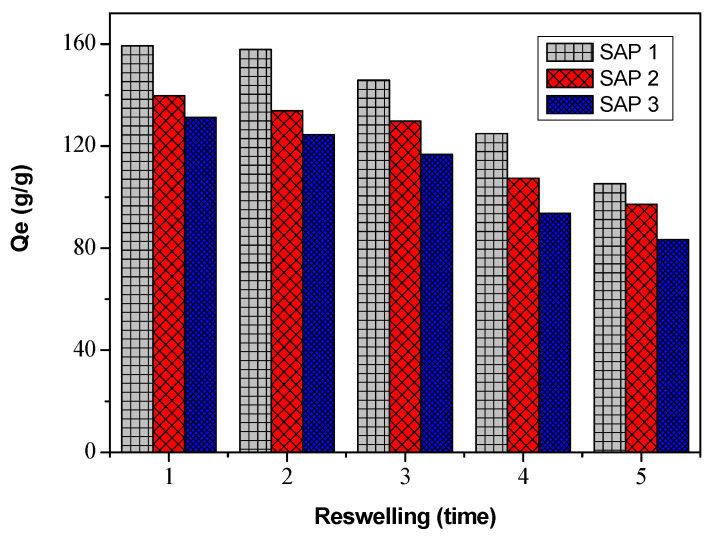
Superabsorbent reuse.

**Figure 6 polymers-14-05175-f006:**
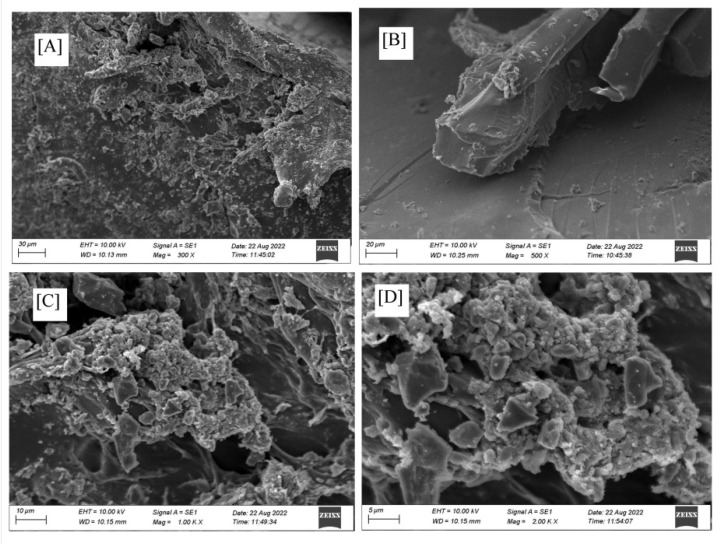
Superabsorbent morphology analysis using SEM with various magnifications (**A**) 300×, (**B**) 500×, (**C**) 1000×, and (**D**) 2000×.

**Figure 7 polymers-14-05175-f007:**
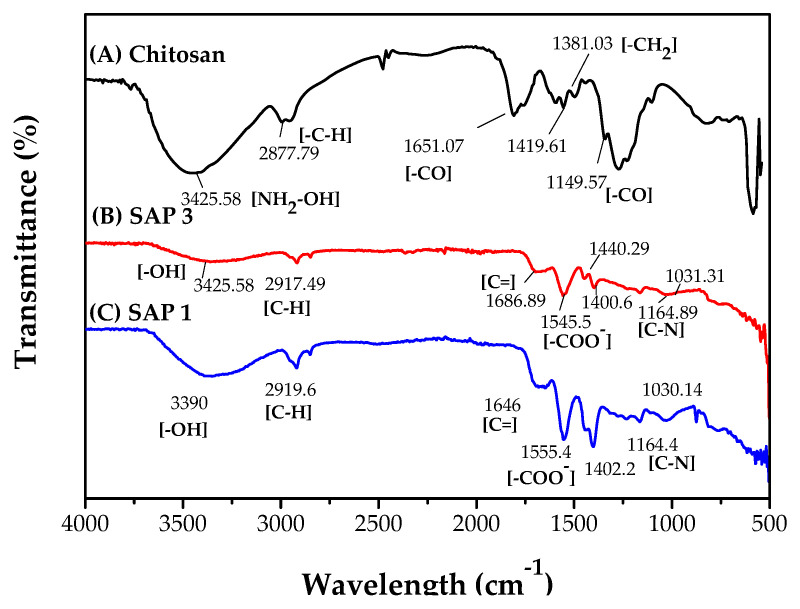
The IR analysis for (**A**) pure chitosan, (**B**) superabsorbent prepared using 0.5% (*w*/*v*) chitosan concentration (SAP 3), and (**C**) superabsorbent using 4% (*w*/*v*) chitosan concentration (SAP 1).

**Figure 8 polymers-14-05175-f008:**
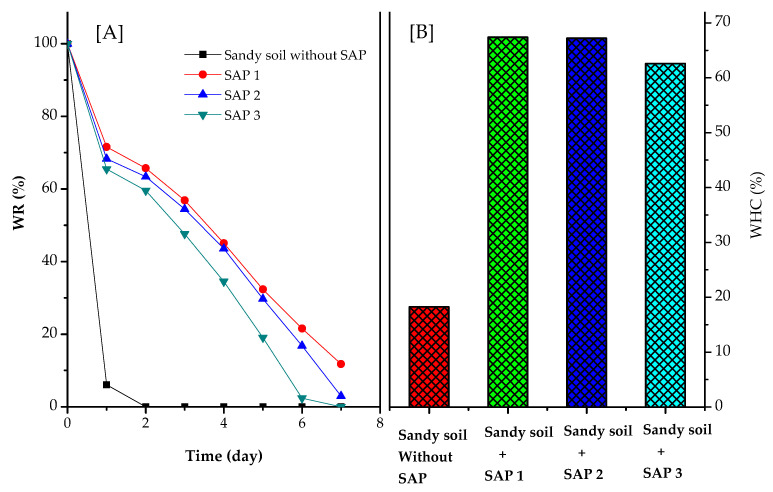
Calculations of (**A**) water holding capacity (WHC) and (**B**) water retention (%) of sandy soil samples with and without the addition of superabsorbent (SAP 1, SAP 2, and SAP 3).

**Figure 9 polymers-14-05175-f009:**
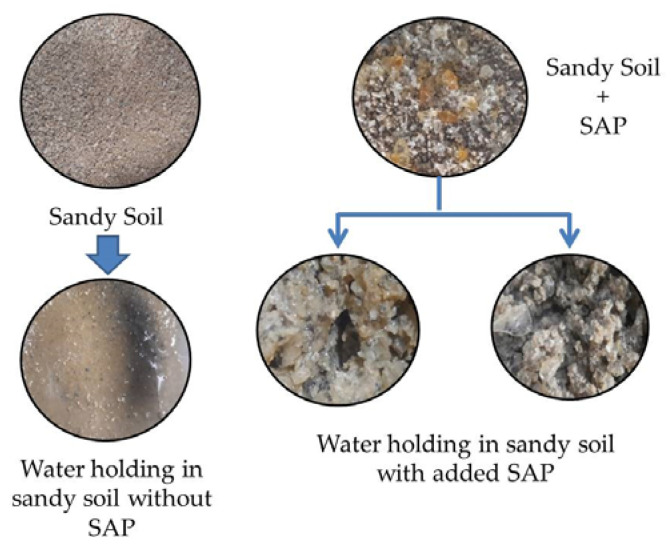
Differences in the water-holding characteristics of sandy soil with and without SAP addition.

**Table 1 polymers-14-05175-t001:** Symbols.

No	Symbol	Definition
1	SAP 1	Acrylic acid superabsorbent grafted with chitosan concentration of 4% (*w*/*v*)
2	SAP 2	Acrylic acid superabsorbent grafted with chitosan concentration of 2% (*w*/*v*)
3	SAP 3	Acrylic acid superabsorbent grafted with chitosan concentration of 0.5% (*w*/*v*)

**Table 2 polymers-14-05175-t002:** Absorption capacity at equilibrium (*Qe*) and rate constants obtained using Swelling Kinetic Models for SAP 1, SAP 2, and SAP 3.

Formulation	Models	Absorption Capacity at Equilibrium (*Qe*) (g/g)	Rate Constants (*k*) [g/(g·min)]	R^2^
SAP 1	Pseudo-first-order	149.941	0.04224	0.990
	Pseudo-second-order	158.814	0.00043	0.996
SAP 2	Pseudo-first-order	135.591	0.04785	0.998
	Pseudo-second-order	142.350	0.00059	0.993
SAP 3	Pseudo-first-order	120.943	0.03850	0.982
	Pseudo-second-order	128.313	0.00044	0.990

## Data Availability

All data is contained in the article.

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
