# Peer review of "Chitosan-Graft-Poly(acrylic acid) Superabsorbent’s Water Holding in Sandy Soils and Its Application in Agriculture"

_polymers, 2022, doi:10.3390/polym14235175_

Round 1
Reviewer 1 Report
The methodology lacks the characterization of the sand used since it does not give values for the amount of nitrogen, carbon, or minerals present—these influence the gelation of this with the gel.
As it was, the residual monomer was removed entirely from the network.
In the reaction mechanism, it remains to be clarified; because it is a polymerization in chitosan, the reaction rate and reactivity of acrylic acid are much faster, so there is a network, not a polymer injected into chitosan. If it were as explained, a proton and carbon NMR are not presented to find out if this graft is.
In the bibliography consulted in the mechanism discussion, the authors present a different mechanism in the case of Shixin Fang et. Col. (2019), which starts with chloroacetyl chitosan and then reacts with acrylic acid.
The written mechanism needs to be corrected, and please take care in the chitosan unit must be written.
Where it says that the polymerization begins is said to be polylactic acid; see figure 1.
The written mechanism needs to be corrected, and please take care in the chitosan unit must be written.
Where it says that the polymerization begins is said to be polylactic acid; see figure 1.
The author of Shixin Fang et al. (2019) shows that it is a network.
In the gelation study, it is necessary to put the percentage of error and/or ANOVA, and in methodology, say how many repetitions were made.
In paragraphs 225-226, it is somewhat confusing, it should be explained what really happens, if there is a greater amount of Chitosan in the network, with poly (acrylic acid). The effect of chitosan of only 4%, 25, 0.5%, is not significant, more chitosan could be added to have more of this effect and observe the effect well.
In figures 2 and 4 you must put the error, to validate this.
The kinetic study must be at different temperatures and pH, to know its behavior to ions.
The effect of the gel on the sand must first be characterized to observe the effect of the gel, as the investigations mentioned in the paper did.
On the other hand, the polluting effect of the gel, there are residual monomers from the synthesis.
The author of Shixin Fang et al. (2019) shows that it is a network.
In the reaction mechanism, it remains to be clarified; because it is a polymerization in chitosan, the reaction rate and reactivity of acrylic acid are much faster, so there is a network, not a polymer injected into chitosan. If it were as explained, a proton and carbon NMR are not presented to find out if this graft is.
In the bibliography consulted in the mechanism discussion, the authors present a different mechanism in the case of Shixin Fang et. Col. (2019), which starts with chloroacetyl chitosan and then reacts with acrylic acid.
The written mechanism needs to be corrected, and please take care in the chitosan unit must be written.
Where it says that the polymerization begins is said to be polylactic acid; see figure 1.

Author Response
Dear reviewer 1.
We really thank you for all the comments and suggestions to make this paper of high quality and worthy of publication. we make every effort to improve this article according to the suggestions given. we cannot add some suggestions because we did not collect data as suggested. We hope that this article can be published in the Journal "Polymers". thank you for all the suggestions that have been given for our article.

Reviewer 2 Report
Line 4. “Jayanudin” Is the name of the first author correct?
Lines 83-84. “The neutralized acrylic acid was then allowed to 83 stand for 24 hours.” What is the purpose of this conditioning of a compound solution prone to polymerization?
Line 88. “then 0.1 g of N, N'-methylenebisacrylamide (MBA) was added” How is uniform distribution of the reagent in the reaction mixture ensured?
Line 320. Header formatting fails.
Author Response
Dear reviewer 2.
We are very grateful for the advice that has been given. We try our best to improve our articles as suggested by reviewers. We hope that this article can be accepted in the "Polymers". Thank you

Round 2
Reviewer 1 Report
x